# Biogeochemical Cycles in Plant–Soil Systems: Significance for Agriculture, Interconnections, and Anthropogenic Disruptions

**DOI:** 10.3390/biology14040433

**Published:** 2025-04-17

**Authors:** Wajid Zaman, Asma Ayaz, Daniel Puppe

**Affiliations:** 1Department of Life Sciences, Yeungnam University, Gyeongsan 38541, Republic of Korea; wajidzaman@yu.ac.kr; 2Faculty of Sports Science, Ningbo University, Ningbo 315211, China; asma@nbu.edu.cn; 3Leibniz Centre for Agricultural Landscape Research (ZALF), 15374 Müncheberg, Germany

**Keywords:** ecosystem services, global change, nutrient dynamics, carbon sequestration, microbial communities, ecosystem resilience, sustainable agriculture

## Abstract

Biogeochemical cycling is essential for maintaining the balance of nutrients/elements in ecosystems. Carbon, nitrogen, phosphorus, sulfur, and silicon cycles facilitate nutrient/element transfer and storage ensuring that plants and soil organisms receive the components necessary for life. This review provides a comprehensive overview of interconnected biogeochemical cycles in terrestrial ecosystems with a focus on agricultural plant–soil systems. The review aims to explore underlying mechanisms and interactions and to derive implications for ecosystem dynamics and services. Moreover, the negative impacts of human activities on biogeochemical cycles are addressed, and mitigation strategies and sustainable management practices are presented. Key findings reveal that while each cycle operates through distinct processes, their coupling is essential for maintaining ecosystem balance and productivity. The disruptions caused by human activities like industrial agriculture or deforestation pose significant challenges to the stability of these cycles. At the same time, advancements in technology, particularly artificial intelligence, remote sensing, and soil health monitoring, offer transformative opportunities to study and manage these cycles with greater precision and efficiency. These innovations can help to identify hotspots of nutrient/element deficiencies or disruptions, predict ecosystem responses to environmental changes, and guide future research and policy development regarding sustainable management practices.

## 1. Introduction

Biogeochemical cycles are fundamental processes that regulate the flow of essential nutrients (like carbon (C), nitrogen (N), phosphorus (P), and sulfur (S)) as well as beneficial elements (like silicon (Si)) through the biosphere, lithosphere, hydrosphere, and atmosphere. These cycles are crucial for sustaining life on Earth by ensuring the availability of nutrients/elements required for growth, reproduction, and ecosystem functioning [1]. In plant–soil systems, biogeochemical cycling plays a pivotal role by mediating nutrient/element exchange between plants and soil, influencing the composition and productivity of ecosystems. This interplay of biotic and abiotic components underscores the intricate relationships that define ecological stability and resilience [2]. The concept of biogeochemical cycles emerged as scientists sought to understand how nutrients/elements move through natural systems, tracing elemental pathways across different environmental compartments and uncovering their interactions with biological processes [3].

The historical context of research into biogeochemical cycling reveals a growing recognition of its complexity and ecological importance [4]. Early studies primarily focused on individual cycles, such as the C and N cycles, examining their contributions to soil fertility and atmospheric regulation [5]. Over time, advances in molecular biology, isotopic techniques, and remote sensing expanded our ability to investigate these cycles on multiple scales, from microbial processes in soil to global fluxes of C and N. The increasing availability of data has enabled the identification of interconnected feedback loops and the recognition of human-induced disruptions, such as deforestation and industrial agriculture, which have altered natural nutrient/element flows [6,7,8].

Biogeochemical cycling is indispensable for maintaining ecosystem health, as it supports key functions like primary productivity, nutrient/element recycling, and soil formation [4,9,10]. The nutrient/element exchange facilitated by these cycles sustains plant growth, which, in turn, influences the structure and function of terrestrial and aquatic ecosystems. This relationship is exemplified by the interdependence between plants, soil, and microbial communities [11]. Plants absorb nutrients/elements from the soil to fuel their metabolic activities, while microbial communities decompose organic matter, releasing nutrients/elements back into the soil in forms accessible to plants. This dynamic feedback system ensures the continuity of life-sustaining processes, highlighting the importance of conserving biogeochemical integrity in the face of environmental changes [12].

Understanding the significance of biogeochemical cycles extends beyond their ecological implications, as they are also central to addressing global challenges such as climate change, food security, and biodiversity loss [13]. For instance, the C cycle’s role in regulating atmospheric carbon dioxide (CO_2_) levels is critical for mitigating climate change [14]. Similarly, the N and P cycles influence agricultural productivity and water quality, underscoring the need for sustainable nutrient management practices. The Si cycle is closely linked to the C cycle on a global scale and the positive effects of Si on plant performance, crop production, and ecosystem functioning are well-documented in the literature [15,16,17]. Biogeochemical cycles also contribute to the resilience of ecosystems by supporting species diversity and enabling ecosystems to recover from disturbances [18,19]. Therefore, any imbalance in biogeochemical cycling, whether due to natural events or human activities, poses significant risks to ecosystem stability and human well-being [20,21].

This review provides a comprehensive overview of interconnected biogeochemical cycles in terrestrial ecosystems with a focus on agricultural plant–soil systems. The review aims to explore underlying mechanisms and interactions and to derive implications for ecosystem dynamics and services. By delving into the details of selected nutrient/element, i.e., C, N, P, S, and Si, cycles, this review elucidates their roles in maintaining ecosystem health and how they are connected. Additionally, this review seeks to address the negative impacts of human activities on biogeochemical cycles, offering insights into mitigation strategies and sustainable management practices. Through a synthesis of existing knowledge and identification of research gaps, this review finally aims to foster a deeper understanding of biogeochemical C, N, P, S, and Si cycling, guiding future research and policy development in ecosystem conservation and management.

## 2. A Short Overview of Carbon, Nitrogen, Phosphorus, Sulfur, and Silicon Cycles in Plant–Soil Systems

Biogeochemical cycles are essential for maintaining the balance of nutrients/elements in ecosystems, especially within plant–soil systems where these cycles drive the availability of critical elements [22,23]. C, N, P, S, and Si cycles comprise key processes that facilitate nutrient/element transfer and storage, ensuring that plants and soil organisms receive the necessary components for growth, energy production, and structural development [2,24]. Each of these cycles operates through unique mechanisms and pathways, interacting with other cycles in complex ways that underscore the intricate relationships between biotic and abiotic factors [25,26]. The following sections provide a short examination of each cycle discussing selected underlying mechanisms, its significance in plant nutrition, and specific challenges. For reasons of clarity and consistency, every section is divided into three subsections. In each first subsection, a short nutrient/element profile is given, which briefly summarizes information on the element’s natural plant availability, its significance in plant nutrition, molecular transformation processes, element limitations and management, and consequences of land-use for the biogeochemical cycle of this element. In the second and third subsections, selected aspects from the first subsection are then presented in more detail.

### 2.1. Carbon Cycle

#### 2.1.1. Carbon Profile

(i) Natural plant availability: While the vast majority of inorganic C in terrestrial plants is sequestered from atmospheric CO_2_ by photosynthesis, C can also be absorbed (in the form of CO_2_, H_2_CO_3_ (carbonic acid), HCO_3_^−^ (bicarbonate), or CO_3_^2−^ (carbonate), depending on soil solution pH) and fixed by roots to some extent [27,28]. (ii) Significance in plant nutrition: Essential. Component of all organic compounds, which represent the basis of all known life. (iii) Transformation processes: Photosynthesis, decomposition, and respiration (see Section 2.1.2). (iv) Limitations and management: Not limited. C sequestration in soil represents a promising pathway of climate change mitigation (see Section 2.1.2). (v) Consequences of land-use for biogeochemical C cycling: increased CO_2_ emissions and decreased C sequestration (see Section 2.1.3).

#### 2.1.2. Carbon Respiration and Sequestration in Soils

The C cycle is fundamental to plant–soil systems, mediating the storage and transfer of C between the atmosphere, soil, and vegetation [29]. While soil respiration by plant roots, microorganisms (bacteria, fungi), and soil animals is a key source of global CO_2_ release, CO_2_ is also sequestered in soils [30]. On a global scale, the uptake of C from the atmosphere by land plant photosynthesis and the C release rates by soil respiration are about the same size [31]. However, heterotrophic soil respiration has been found to substantially increase, driven by global warming, which might have severe consequences for the global C budget on a short time scale (i.e., up to hundreds of years; on a long time scale, the C budget is assumed to be in equilibrium) [32]. C sequestration in soil occurs through two primary biological mechanisms: photosynthesis and decomposition. During photosynthesis, plants absorb CO_2_ from the atmosphere, converting it into organic compounds that are stored in plant tissues [33]. When plants and organic materials decompose, microorganisms break down these compounds, integrating C into soil organic matter (SOM) [34]. This SOM, along with humus, serves as a major reservoir for C, enhancing soil fertility and water retention [35]. Agricultural practices to enhance biological C sequestration in soils comprise, e.g., conservation agriculture, cover cropping, or the recycling of biomass (see Section 4.2 for more details). On a multimillion-year time scale, atmospheric CO_2_ concentrations are largely controlled by the removal of CO_2_ from the atmosphere through chemical soil weathering [36]. Enhancing this chemical weathering, e.g., by liming, might be another promising approach to increase the uptake of CO_2_ from the atmosphere on a global scale [37].

#### 2.1.3. Impact of Land-Use on Carbon Cycling

Land-use significantly disrupts C dynamics in the soil. Deforestation reduces C sequestration by removing plant biomass that stores C, while agricultural practices such as tilling accelerate the decomposition of organic matter, leading to increased C release [38]. Urbanization exacerbates these effects by reducing vegetative cover and compacting soils, further limiting their capacity to store C [39]. All these human activities highly impact the storage of C within plant biomass with consequences for C sequestration in soils and finally the release of C to the atmosphere (see Section 4.2 for a detailed overview). Key processes of C movement in plant–soil systems, i.e., C sequestration, storage, and release, and the impact of land-use are summarized in Figure 1.

### 2.2. Nitrogen Cycle

#### 2.2.1. Nitrogen Profile

(i) Natural plant availability: While the vast majority of plant available N in soil originates from the conversion of atmospheric dinitrogen (N_2_) into ammonia (NH_3_) by bacteria (biological N fixation), some atmospheric N_2_ is abiologically fixed by lightning (see Section 2.2.2). (ii) Significance in plant nutrition: Essential. As it is a component of diverse metabolites and structural compounds like proteins, nucleic acids, chlorophylls, phytohormones, and secondary metabolites, N plays a vital role in plant physiology. (iii) Transformation processes: N fixation and nitrification/denitrification (see Section 2.2.2). (iv) Limitations and management: Not limited in natural ecosystems, where the N cycle is in equilibrium. However, in agricultural plant–soil systems, the harvest-related substantial annual N losses have to be compensated via N fertilization (see Section 2.2.3). (v) Consequences of land-use for biogeochemical N cycling: emission of the greenhouse gas nitrous oxide (N_2_O), deterioration of soil health, and eutrophication (see Section 2.2.3).

#### 2.2.2. Nitrogen Fixation, Nitrification, and Denitrification

The N cycle plays a critical role in plant–soil systems by converting inert atmospheric N into biologically available forms [40]. N fixation is primarily achieved by symbiotic bacteria such as *Rhizobium* (associated with leguminous plants) and free-living bacteria such as *Azotobacter*. These microorganisms transform atmospheric N_2_ into NH_3_, which plants can absorb [41]. Abiotic processes, including lightning and industrial fixation, also contribute to N availability, albeit to a lesser extent [42]. Lightning provides the energy to produce NO_x_ (nitrogen oxides) in a reaction of N_2_ and oxygen in the atmosphere [43]. When NO_x_ cools down afterwards it further reacts with oxygen forming nitrogen dioxide (NO_2_), which in turn is converted to nitric acid (HNO_3_) during chemical reactions with ozone and water [44]. In soil, this acid finally makes NO_3_^−^ (nitrate), a form readily absorbed by plants [45]. Following N fixation, nitrification and denitrification are key processes that regulate the transformation and movement of N in soil [46]. Nitrification involves the microbial oxidation of ammonia to nitrite (NO_2_^−^) and subsequently to plant available NO_3_^−^. Conversely, denitrification converts nitrate into gaseous N forms (N_2_ or N_2_O), releasing it back into the atmosphere [47].

#### 2.2.3. Nitrogen Supply in Agricultural Plant–Soil Systems

On the one hand, intensive N fertilization has led to improved food supply, because significantly more crops can be grown on a limited area of land as N is the main yield-determining nutrient. On the other hand, the oversupply with reactive N (i.e., all N compounds except for N_2_) in many agricultural plant–soil systems has resulted in ecological threats like the emission of the greenhouse gas N_2_O, deterioration of soil health, or eutrophication (see Section 5.1.) [48,49]. Annual harvest-related N removal from agricultural plant–soil systems has to be replaced to maintain productivity, because there is no potentially available N in the rocks from which most soils developed. To ensure maximal crop yields and minimal N losses in any agricultural plant–soil system, the N fertilization rate should be tailored to the N demand of a specific crop [50]. In this context, synthetic fertilizers are most often the means of choice [51], especially in intensive agricultural systems, because they are readily available, easy to transport and to apply, and relatively cost-effective [52]. During industrial N fertilizer production, N_2_ is fixed in the Haber–Bosch process, which uses hydrogen (H_2_) from natural gas (CH_4_) to react with N_2_ under high temperature and pressure to form NH_3_. Apart from synthetic N fertilizers, there are numerous other N sources used in agricultural plant–soil systems, for example, manure or compost. Table 1 provides a comparative overview of N sources, highlighting their relative contributions to soil fertility and crop productivity.

### 2.3. Phosphorus Cycle

#### 2.3.1. Phosphorus Profile

(i) Natural plant availability: Only inorganic orthophosphates in soils (i.e., H_2_PO_4_^−^ and HPO_4_^2−^) are plant available. (ii) Significance in plant nutrition: Essential. P plays a major role in the structural framework of deoxyribonucleic acid (DNA) and ribonucleic acid (RNA), is used for the transport of cellular energy with adenosine triphosphate (ATP), and represents a main structural component of cell membranes in the form of phospholipids. (iii) Transformation processes: Weathering and mineralization (see Section 2.3.2). (iv) Limitations and management: Global P resources are limited, which is why a sustainable use of P fertilizers is urgently needed in future cropping systems (see Section 2.3.3). (v) Consequences of land-use for biogeochemical P cycling: deterioration of soil health and eutrophication (see Section 2.3.3).

#### 2.3.2. Availability of Phosphorus in Soils

Unlike N and C cycles, the P cycle lacks a significant gaseous phase, relying primarily on the weathering of phosphate-bearing rocks to release P into the soil [61]. Once in the soil, P undergoes mineralization, where organic P is converted into inorganic forms accessible to plants. However, P availability is often limited by its strong affinity for soil particles, which immobilizes it in forms unavailable for uptake [62]. The negatively charged phosphate ions are, for example, adsorbed on positively charged soil constituents such as iron (Fe) and aluminum (Al) oxides, carboxyl groups (organic matter), or silanol groups (clay minerals). Moreover, phosphates are immobilized during mineral precipitation in combination with metals such as calcium (Ca), Fe, and Al [63,64]. Soil amendments such as biochar, rock phosphate, and phosphate-solubilizing microorganisms enhance P accessibility. These interventions are particularly important in agricultural systems where P deficiencies can limit crop yields [30]. Figure 2 illustrates the dynamics of P cycling in plant–soil systems in relation to abiotic and biotic factors.

#### 2.3.3. Global P Resources and Strategies to Reduce the Need for Synthetic P Fertilizers in Agricultural Plant–Soil Systems

Synthetic P fertilizers are mainly manufactured from mined rock phosphate, which is a geographically restricted and finite resource. Currently, China, Morocco, and the United States are the main phosphate producers worldwide, accounting for about 70% of the total global production [65]. The largest phosphate resources on Earth can be found in the Western Phosphate Fields (United States) and the Mediterranean Sedimentary Phosphate Province (e.g., Morocco, Spain, Algeria, Tunisia) with deposits that are estimated to last for more than 1000 years [66]. Despite these huge resources, factors like growing P demand, geopolitical constraints, increasing prices, and ecological awareness against the background of global change call for a sustainable management of global P resources [67]. In this context, agriculture plays a pivotal role regarding the demand for synthetic P fertilizers as most global phosphates are consumed during the production of these fertilizers. The overuse of synthetic P fertilizers results in a deterioration of soil health and P runoff from agricultural lands to water bodies, where it causes eutrophication (see Section 5.1.). To avoid P fertilizer overuse, farmers should assess the P status (i.e., P plant availability) of their soils as well as specific crop P demands before determining P fertilizer rates. In this context, precision agriculture [68,69] and the use of bio-inoculants (bacteria and fungi) [70] represent promising strategies to increase the P use efficiency of crops.

### 2.4. Sulfur Cycle

#### 2.4.1. Sulfur Profile

(i) Natural plant availability: Sulfate (SO_4_^2−^) as the most common inorganic form of S in soils is readily available to plants. Moreover, atmospheric hydrogen sulfide (H_2_S) and sulfur dioxide (SO_2_) can also be used by plants as a source of S, if root S uptake is limited [71]. (ii) Significance in plant nutrition: Essential. As S is a component of several specific biomolecules (e.g., amino acids and cofactors), S plays a crucial role in biochemical functioning of all living organisms. (iii) Transformation processes: Weathering, mineralization, and inorganic transformations (see Section 2.4.2). (iv) Limitations and management: As SO_4_^2−^ contents in soils greatly vary and intensified agriculture leads to substantial S losses, the need for S fertilization is strongly increasing (see Section 2.4.3). (v) Consequences of land-use for biogeochemical S cycling: acidification of terrestrial and aquatic ecosystems and air pollution (see Section 2.4.3).

#### 2.4.2. Sulfur Transformation Processes

S can be abundantly found in rocks and soil minerals. Rock and mineral weathering, geothermal vents, and volcanic eruptions contribute to the release of S into the environment. Human activities like fossil fuel combustion and agriculture largely influence the biogeochemical S cycle [72]. The content of SO_4_^2−^ in soils greatly varies in relation to factors like climate, vegetation, soil processes (e.g., leaching, adsorption–desorption, precipitation, and oxidation-reduction reactions), and soil microbial activity. S cycling in plant–soil systems involves the transformation of S compounds through microbial and chemical processes [73]. In aerobic conditions, microorganisms oxidize S compounds to SO_4_^2−^, the most plant-available form of S [74]. In anaerobic environments, sulfate-reducing bacteria convert sulfate into H_2_S, completing the S cycle [75].

#### 2.4.3. Sulfur Management in Agricultural Plant–Soil Systems

S is vital for plant nutrition, as it forms the backbone of essential amino acids like cysteine and methionine. These amino acids are critical for protein synthesis and enzymatic functions. S inputs to agricultural plant–soil systems in the 19th and 20th centuries were mainly in the form of atmospheric wet (“acid rain”) and dry depositions, which were driven by industrial SO_2_ emissions and had negative environmental impacts, i.e., acidification of terrestrial and aquatic ecosystems and air pollution [76]. Due to stricter emission regulations in many countries worldwide, industrial S emissions are decreasing [72]. However, this emission decrease in turn has increased S deficiencies in soils, and thus the need for S fertilization in intensified cropping systems with environmental consequences that might be comparable to the ones described above [77]. This need for S fertilization is exacerbated by S losses from agricultural soils driven by, e.g., topsoil erosion and leaching [78,79]. To address S deficiencies in agricultural systems, S-rich fertilizers and soil amendments are commonly applied [80]. Table 2 summarizes various S-related soil amendments, their sources, and their impacts on plant health.

### 2.5. Silicon Cycle

#### 2.5.1. Silicon Profile

(i) Natural plant availability: Only dissolved silica (i.e., silicic acid, H_4_SiO_4_) in soils is plant available. (ii) Significance in plant nutrition: Beneficial. As silica accumulation in higher plants has been found to enhance their resistance against abiotic and biotic stress, Si is considered as quasi-essential for plants, especially for plant species from the grass family (Poaceae) [83]. (iii) Transformation processes: Weathering and biosilicification (see Section 2.5.2). (iv) Limitations and management: Substantial Si losses in agricultural plant–soil systems due to annual harvest-related Si exports, especially driven by cereal crop harvesting (see Section 2.5.3). (v) Consequences of land-use for biogeochemical Si cycling: anthropogenic desilication and changes in land–ocean Si fluxes (see Section 2.5.3).

#### 2.5.2. Biosilicification and Its Role in Silicon Cycling

Si has been found to be a beneficial element for plants, enhancing their resistance to stress, and thus positively affecting plant performance and ecosystem functioning, especially under stress conditions [9,84,85]. The original source of bioavailable Si (i.e., dissolved silica in the form of H_4_SiO_4_) on a geological time scale is mineral weathering. In soils, H_4_SiO_4_ follows various pathways including abiotic and biotic ones. While abiotic pathways comprise (i) leaching influenced by rainfall and irrigation and (ii) immobilization through soil processes like adsorption, precipitation, and complexation, the biotic pathway is represented by the uptake of H_4_SiO_4_ by living organisms [86]. The process by which inorganic H_4_SiO_4_ is utilized by plants, protists, and animals to create biogenic silica (BSi), i.e., amorphous hydrated silica (SiO_2_∙*n*H_2_O), is known as biosilicification [87]. As BSi is much more soluble compared to silicate minerals, biosilicification and BSi dissolution regulate H_4_SiO_4_ concentrations in soils over biological time scales [88,89].

In fact, various organisms can accumulate BSi. However, while particular attention has been paid to phytogenic (plants) silica pools [90,91,92], less is known about protozoic (testate amoebae), protophytic (diatoms), zoogenic (sponges), fungal (fungi), and bacterial (bacteria) Si pools in soils (Table 3) [93]. While protozoic Si pools in forest soils are relatively small compared to phytogenic ones, biosilicification rates of testate amoebae can equal or even exceed the annual Si uptake rates of trees in terrestrial ecosystems [94,95]. As testate amoebae belong to the earliest colonizers of new areas [96], they might also play a crucial role in establishing biological Si cycling in initial ecosystems [95,97]. Protophytic and zoogenic Si pools in soils were quantified in few studies, but there are no quantitative data for corresponding biosilicification rates by diatoms and sponges in terrestrial ecosystems, respectively [98,99]. Regarding fungal and bacterial biosilicification, there is no quantitative data (Si pools, annual Si uptake rates) in the literature to the best of our knowledge.

#### 2.5.3. The Threat of Anthropogenic Desilication and How to Prevent It

Human activities like deforestation and damming have profound impacts on Si cycling in terrestrial ecosystems with consequences for Si fluxes from land to oceans, and thus for the global biogeochemical Si cycle (see Section 5.1). Moreover, annual Si exports by crop harvesting have been recognized to lead to substantial Si losses in agricultural plant–soil systems known as anthropogenic desilication [107,108,109]. Certain agricultural practices may enhance Si availability (H_4_SiO_4_) in soils like the use of prescribed burning [110], the application of Si-rich fertilizers and/or amorphous silica [17,111], the application of biochar [112], the recycling of crop residues [113,114], and liming [115]. However, further research is needed to determine how much of the released H_4_SiO_4_ is (i) taken up by plants, (ii) immobilized through adsorption and complexation processes in the soil, or (iii) lost through leaching.

All these practices can prevent anthropogenic desilication and ensure a sufficient Si supply for plants leading to an improved crop stress resilience [85,111]. Additionally, practices like biochar application or crop residue recycling can promote C sequestration in agricultural soils by enhancing weathering (e.g., using silicate rock powders), stabilizing soil organic C (e.g., through crop residue recycling or biochar application), and increasing phytolith-occluded C storage (e.g., by enhancing cereal crop production and corresponding residue recycling) [116,117].

## 3. Interactions Between Biogeochemical Cycles

Biogeochemical cycles rarely operate in isolation. Instead, they are interconnected, with processes in one cycle influencing and being influenced by others. This interconnectedness enhances the complexity of nutrient/element dynamics within plant–soil systems, as each cycle interacts with others to maintain ecosystem stability and productivity [118]. The coupling between cycles such as C, N, and P, along with the pivotal role played by microbial communities, creates a dynamic feedback network. These interactions determine nutrient/element availability, ecosystem resilience, and plant productivity [119]. A deeper understanding of these interactions is critical for managing ecosystem sustainably, particularly in the context of anthropogenic disruptions like land-use change and climate variability [120].

### 3.1. Underlying Mechanisms

The coupling of biogeochemical cycles refers to the feedback mechanisms that connect processes within different cycles [121]. For instance, the C, N, and P cycles are intricately linked through plant growth, microbial activity, and soil processes [122]. Photosynthesis, a key process in the C cycle, requires N for chlorophyll synthesis and P for energy transfer via ATP molecules [123]. N fixation, a major process in the N cycle, is often influenced by the availability of organic C, which serves as an energy source for N-fixing bacteria [124]. Similarly, P availability impacts microbial activity and root exudates, further influencing C dynamics in the soil [125]. Since S is also an essential element for organisms (e.g., as a constituent of many proteins and cofactors), the S cycle is closely linked to the other biogeochemical cycles by biological productivity as well. The global Si and C cycles are closely linked by weathering processes and diatom growth in the oceans [126,127]. While soil weathering principally removes CO_2_ from the atmosphere, tectonic activities can expose ancient organic C that was sequestered from the atmosphere over millions of years and causes CO_2_ release when this organic C is oxidized [128]. Finally, volcanic eruptions worldwide emit gases rich in C and S, i.e., mainly CO_2_ and SO_2_, but also N species such as NO_x_ into the atmosphere [129,130], while volcanic ash is rich in Si (volcanic glass) and also contains certain amounts of P [131,132]. Gaseous S and N species in the atmosphere (i) are transformed into aerosol acid particles (SO_4_^2−^ and NO_x_) that eventually fall to the ground in the form of dry depositions or (ii) react with water, oxygen, and other chemicals to form sulfuric (H_2_SO_4_) and nitric (HNO_3_) acids, which are then deposited via precipitation (wet depositions) [130,133].

However, at this point, it should be noted that the processes stated above naturally represent only some examples of biogeochemical cycle connections, because the elements C, N, P, S, and Si are quite abundant in the Earth’s system, and corresponding interactions between the Earth’s spheres (i.e., the litho-, hydro-, bio-, and atmosphere) are just too complex to be depicted in full detail. While all these natural processes are globally in equilibrium over a long time scale, human activities substantially disturb these cycles over a short time scale (i.e., decades to hundreds of years) with far reaching consequences for element cycling and availability in plant–soil systems, and thus for ecosystem resilience and plant productivity. Figure 3 provides an overview of natural key processes (e.g., photosynthesis, N fixation, and rock weathering) and human disturbances, which are driving nutrient/element exchange between the Earth’s spheres.

The feedback mechanisms of biogeochemical cycles have profound implications for plant productivity and ecosystem functioning. When nutrient/element availability is balanced across these cycles, plants can optimize growth and yield. However, imbalances, such as N or P limitations, can disrupt C sequestration and reduce ecosystem productivity [134]. For example, excess N from fertilizers can lead to P depletion, altering microbial communities and reducing soil health (see also Section 4.3) [135].

### 3.2. Role of Microbial Communities

Microbial communities are central to the interactions between biogeochemical cycles, acting as catalysts for nutrient/element transformations and exchange. These communities engage in symbiotic relationships with plants, facilitating nutrient/element acquisition and cycling [136]. For example, N-fixing bacteria like *Rhizobium* form symbiotic associations with leguminous plants, converting atmospheric N into ammonia, which the plants can use [137]. Similarly, mycorrhizal fungi establish mutualistic relationships with plant roots, enhancing P uptake in exchange for C from the plant [138]. Modern molecular biological techniques are well-suited to understand the underlying processes of the cross-talk between plants and soil microorganisms, and thus to regulate these interactions for the benefit of plant nutrition [139,140,141,142].

The enzymatic processes carried out by microbes are critical for driving nutrient/element availability and cycling [143]. Microbial enzymes decompose organic matter, releasing N, P, and C compounds into the soil [144]. Specific microbial species also mediate processes such as nitrification, denitrification, and P solubilization, bridging the gaps between different biogeochemical cycles. Furthermore, certain bacteria (e.g., *Proteus mirabilis*) and fungi can enhance the dissolution of silica via acidic metabolites (bio-weathering) [87]. Diatoms might also play a role in enhancing bio-weathering [145], while there is no information on this aspect regarding testate amoebae.

These activities not only sustain plant growth but also regulate nutrient/element losses and emissions, such as N leaching or nitrous oxide release [146]. To highlight their contributions, Table 4 presents a summary of key microbial species involved in nutrient cycling, their functional roles, and the processes they drive. For instance, *Azotobacter* and *Bradyrhizobium* are pivotal in N fixation, while phosphate-solubilizing bacteria like *Pseudomonas* and *Bacillus* enhance P availability. By bridging cycles, microbes play a crucial role in maintaining nutrient balance and ensuring the resilience of plant–soil systems [147]. The intricate interactions between biogeochemical cycles, facilitated by microbial communities, underscore the complexity of nutrient dynamics in ecosystems. These interactions not only regulate the flow of essential nutrients but also enhance ecosystem productivity and resilience [148].

## 4. Implications for Ecosystem Services

Biogeochemical cycles are not just fundamental processes for nutrient/element dynamics but also pivotal drivers of ecosystem services, which encompass the benefits that ecosystems provide to humanity. These cycles directly influence soil fertility, C sequestration, climate mitigation, biodiversity, and ecosystem resilience [160]. By mediating the availability of essential nutrients and beneficial elements, they support primary productivity, regulate greenhouse gas fluxes, and maintain ecosystem stability under varying environmental conditions [161]. Understanding and managing the implications of these cycles is crucial for promoting sustainable agricultural practices, enhancing climate resilience, and preserving biodiversity.

### 4.1. Soil Fertility and Crop Production

Soil fertility, a cornerstone of agricultural productivity, is intrinsically tied to the efficiency of biogeochemical cycles. These cycles regulate the availability of macronutrients like N, P, and potassium (K), which are critical for crop growth and yield [162]. Sustainable practices such as crop rotation, cover cropping, and the use of organic fertilizers enhance nutrient cycling, ensuring a steady supply of these nutrients [163]. For example, N fixation by legumes in rotation systems replenishes soil N, reducing the need for synthetic fertilizers and improving soil health [164]. Continuous crop straw recycling has been found to replenish plant available Si in agricultural soils and to a reduce the need for N fertilization rates by about 69% in the long term [113]. Additionally, regular long-term incorporation of crop straw is a promising strategy to alleviate soil erosion [165] and to enhance C sequestration in agricultural soils [116,166]. As dissolved Si and P compete for equivalent adsorption sites in soil, Si supply can also help to release previously plant unavailable phosphates in agricultural soils resulting in a reduced need for P fertilization [111]. Due to the manifold reported positive effects of Si for agricultural soils, Si supply has been suggested as crucial for soil health, especially in plant–soil systems that are prone to drought and soil degradation [167].

The impact of biogeochemical processes on yield is profound. Proper nutrient/element cycling ensures optimal root development, photosynthesis, and energy transfer within plants, translating into higher crop productivity [168]. Conversely, disruptions in these cycles, often caused by intensive farming or soil degradation, can lead to nutrient/element imbalances, reduced yields, and long-term soil infertility [169]. Figure 4 presents pathways of nutrient/element transfer from soil to crops, highlighting the role of sustainable practices in enhancing these processes.

### 4.2. Carbon Sequestration and Climate Mitigation

Soil serves as a major C sink, playing a critical role in mitigating climate change by sequestering atmospheric CO_2_ [170]. Biogeochemical processes such as photosynthesis and the decomposition of organic matter drive soil C storage, with SOM acting as a stable reservoir of C [171]. Enhanced C sequestration not only reduces greenhouse gas concentrations but also improves soil structure, water retention, and fertility, creating a positive feedback loop for agricultural productivity [172].

Strategies for maximizing soil C storage include conservation tillage, agroforestry, and the application of biochar. These practices reduce soil disturbance, increase organic C inputs, and promote microbial activity, which stabilizes C in the soil [173]. Moreover, crop straw recycling has been found a promising strategy to replenish plant available Si (H_4_SiO_4_) and to enhance C sequestration in agricultural soils in the long term [113,114]. In this context, a certain amount of C is stored in phytoliths (i.e., silica bodies formed in terrestrial plants), which are transferred to soil with plant litter. As phytoliths are relatively stable and can remain in soils up to centuries or even millennia, C sequestration via soil phytoliths has been recently discussed as a promising long-term C sink [174,175]. Table 5 provides a comparative analysis of soil management practices, outlining their potential for C sequestration and their co-benefits for soil health and ecosystem services. By adopting such strategies, farmers and land managers can contribute to both climate mitigation and sustainable development goals.

### 4.3. Biodiversity and Ecosystem Resilience

Biodiversity and ecosystem resilience are deeply influenced by the efficiency of nutrient/element cycling within ecosystems [9,180]. Healthy biogeochemical cycles promote plant diversity by ensuring the availability of diverse nutrients/elements needed for different species. This diversity, in turn, supports complex food webs and enhances ecosystem functionality [181]. For example, P cycling facilitates the growth of nutrient-demanding species, while N cycling supports legumes and other N-fixing plants, contributing to species coexistence and diversity [182]. Si uptake by plants might play an important role in influencing plant community assembly and ecosystem structure, thus affecting plant biodiversity patterns and ecosystem functioning as well as resilience [183].

In nutrient-poor soils, resilience mechanisms such as microbial symbioses and adaptive root systems play a crucial role in maintaining ecosystem stability. In this context, changes in the root architecture, the formation of cluster roots, or symbiotic associations with mycorrhizae or N-fixing bacteria represent key processes [184]. These mechanisms allow plants to access limited nutrients, ensuring ecosystem productivity even under stressful conditions [185]. Regarding crop production, the phenotypic characterization of root adaptations allows plant breeders to develop improved cultivars, which can ensure yield stability and nutritional security under global change [186]. Moreover, the application of Si to soils has been found to be a promising strategy to enhance the resilience of soil microbial communities by, e.g., changing soil pH, improving nutrient and water availability, altering root exudation patterns and plant physiology, and stimulating the abundance, diversity, and functional potential of key microbial groups [187]. Si uptake by legumes (Fabaceae) might also promote the symbiotic interactions between N-fixing bacteria (rhizobia) inside the legume root nodules and their host plant [188]. Furthermore, enhancing the biological N fixation by introducing selected, adapted diazotrophic bacteria into agricultural soils might be a promising approach to significantly reduce the high demand for synthetic N fertilizers in future cropping systems [189]. Figure 5 provides an overview of how nutrient/element cycling enhances ecosystem resilience, showing pathways through which biogeochemical cycles support biodiversity and mitigate stress. This resilience is vital for ecosystem recovery from disturbances, such as droughts or human interventions, underscoring the need to conserve these natural processes [190]. The implications of biogeochemical cycles for ecosystem services are far-reaching, influencing food security, climate stability, and biodiversity conservation [191].

## 5. Challenges and Future Perspectives

The intricate functioning of biogeochemical cycles is increasingly under threat from human activities, posing significant challenges to ecosystem stability and the delivery of critical ecosystem services [192]. Addressing these challenges requires a combination of mitigation strategies, technological innovations, and forward-thinking research.

### 5.1. Anthropogenic Impacts on Biogeochemical Cycles

Industrial agriculture and urban development have profoundly disrupted natural biogeochemical cycles, altering nutrient/element flows and ecosystem dynamics [193]. Intensified agriculture often characterized by an overuse of synthetic fertilizers and monocropping has led to imbalances in N, P, and Si cycles with far-reaching consequences [18,109]. In general, a continuous overuse of synthetic fertilizers can decline the contents of SOM, change soil pH, reduce soil fertility as well as microbial activity, and thus result in an overall decrease in agricultural soil quality/health [194,195]. More specifically, excessive N application, for instance, contributes to soil acidification, nitrate leaching, and the release of N_2_O, a potent greenhouse gas [196]. Similarly, P runoff from agricultural lands causes eutrophication in freshwater systems, leading to harmful algal blooms and biodiversity loss. Land-use change (e.g., deforestation) has been found to substantially affect Si pools in terrestrial ecosystems with consequences for Si fluxes from land to ocean [8,197,198]. Further human activities like damming also influence the global biogeochemical Si cycle and thus the growth of various organisms (e.g., diatoms) in aquatic ecosystems, as they need dissolved Si to form their siliceous shells [199,200]. Beyond agriculture, urbanization and industrial activities further exacerbate these issues by increasing C emissions and thus global warming, which in turn has strong effects on global biogeochemical cycles [200,201,202].

Pollution is another major driver of biogeochemical cycle disruption. Chemical pollutants, such as “heavy metals” (metal(loid)s) and pesticides, affect microbial communities and soil health, impairing the efficiency of nutrient cycling processes [203]. Eutrophication, caused by nutrient overloading, particularly N and P, depletes oxygen levels in aquatic systems, creating dead zones and disrupting aquatic ecosystems [204]. Table 6 provides an overview of major anthropogenic threats to biogeochemical cycles, alongside potential mitigation strategies, such as precision agriculture, sustainable land management, and pollution control measures.

### 5.2. Technological Advancements

The advent of advanced technologies offers promising solutions for studying and managing biogeochemical cycles more effectively. Artificial intelligence (AI) and remote sensing technologies have revolutionized our ability to monitor and model nutrient dynamics across scales [210]. AI algorithms can process vast datasets from satellite imagery, ground sensors, and climate models, providing insights into patterns of nutrient flows, hotspots of disruption, and potential areas for intervention. Remote sensing, coupled with geospatial analysis, enables the mapping of nutrient deficiencies and soil health metrics at regional and global scales, facilitating targeted and efficient management strategies [211]. Moreover, remote sensing can detect biomass heterogeneities at landscape scales, offering a promising tool for quantifying lifelike Si or C aboveground plant stocks, which are directly related to soil properties and nutrient/element availability [212,213].

Advancements in soil health monitoring techniques further complement these technological strides. Tools such as automated soil sensors and molecular diagnostics allow for real-time analysis of soil properties, including nutrient content, microbial activity, and organic matter levels. These innovations not only enhance the precision of nutrient management but also help predict the impacts of land-use changes and climate variability on biogeochemical cycles [214,215].

### 5.3. Research Gaps and Future Directions

Despite significant progress in understanding biogeochemical cycles, critical knowledge gaps remain, particularly regarding their interlinkages and responses to global change. For instance, while individual biogeochemical cycles are well-studied, the feedback loops and synergies between these cycles require further investigation. Understanding how disruptions in one cycle affect others, particularly under scenarios of climate change and land-use transformation, is essential for developing holistic management approaches. To mitigate anthropogenic disturbances of biogeochemical cycles, natural resources should be the focus of future research. In this context, closing element cycles in agricultural plant–soil systems to the highest extent possible (e.g., by crop residue recycling), using the potentials of (adapted) soil microorganisms and improved plant cultivars, and applying soil conservation methods might be the means of choice.

Future research agendas must prioritize the integration of multiple disciplines to address underlying complexities. This includes coupling biogeochemical research with ecosystem modeling, socio-economic studies, and policy analysis to develop comprehensive frameworks for sustainable management. Additionally, long-term field experiments and multi-scale studies are needed to capture the variability of biogeochemical processes and their responses to both natural and anthropogenic drivers. There is also a pressing need to explore innovative solutions, such as microbial engineering and biogeochemical restoration, to enhance nutrient/element cycling and mitigate the impacts of human activities. Addressing the challenges facing biogeochemical cycles requires a multi-pronged approach that combines the mitigation of anthropogenic impacts, leveraging technological advancements, and closing critical research gaps.

## 6. Conclusions

The intricate and interdependent nature of biogeochemical cycles forms the backbone of nutrient/element dynamics in plant–soil systems, shaping ecosystem health, productivity, and resilience. This review highlights the critical roles played by the C, N, P, S, and Si cycles, emphasizing their unique mechanisms and interactions. These cycles govern the availability and movement of essential nutrients/beneficial elements, sustaining plant growth, microbial activity, and soil health. Key findings reveal that while each cycle operates through distinct processes, their coupling is essential for maintaining ecosystem balance and productivity. The influence of these cycles extends beyond nutrient/element dynamics, impacting climate regulation, biodiversity conservation, and agricultural sustainability.

The disruptions caused by human activities, including industrial agriculture, deforestation, and pollution, pose significant challenges to the stability of these cycles. Excessive nutrient inputs, eutrophication, and C imbalances have led to ecosystem degradation and reduced functionality, emphasizing the urgent need for mitigation strategies. At the same time, advancements in technology, particularly AI, remote sensing, and soil health monitoring, offer transformative opportunities to study and manage these cycles with greater precision and efficiency. These innovations can help to identify hotspots of nutrient/element deficiencies or disruptions, predict ecosystem responses to environmental changes, and guide sustainable management practices.

The implications of biogeochemical cycles for plant–soil systems underscore the necessity of integrated approaches in both research and practice. The interconnectedness of these cycles demands a holistic perspective that considers their feedback mechanisms, synergies, and cascading effects. Future studies must prioritize interdisciplinary research that bridges biology, geology, chemistry, and environmental science to develop comprehensive models of nutrient/element dynamics. These models should include land-use policies, conservation strategies, and agricultural practices to address current challenges and build resilience against future disruptions.

Plant–soil system sustainability hinges on our ability to understand and manage biogeochemical cycles effectively. By synthesizing knowledge across disciplines, leveraging technological advancements, and implementing integrated approaches, the functionality of these cycles and corresponding ecosystem services can be safeguarded. Such efforts are vital for mitigating the impacts of climate change, enhancing food security, and preserving biodiversity, ensuring a sustainable future for both natural ecosystems and human societies.

## Figures and Tables

**Figure 1 biology-14-00433-f001:**
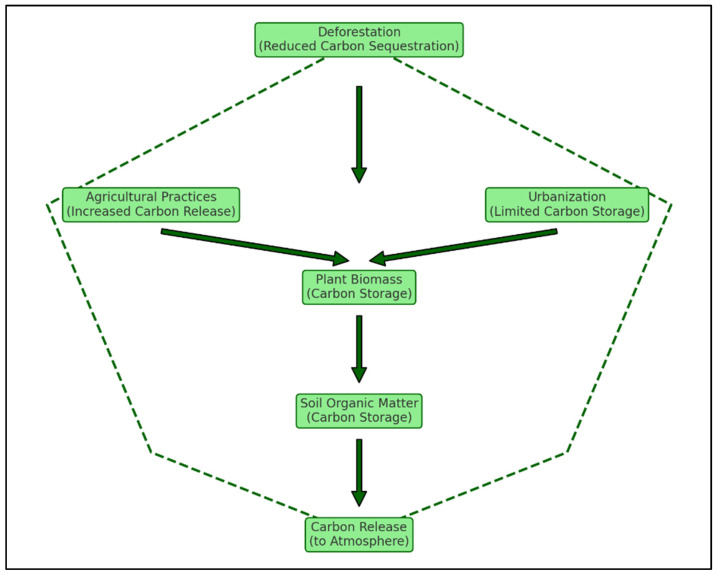
Impact of land-use on C dynamics in plant–soil systems.

**Figure 2 biology-14-00433-f002:**
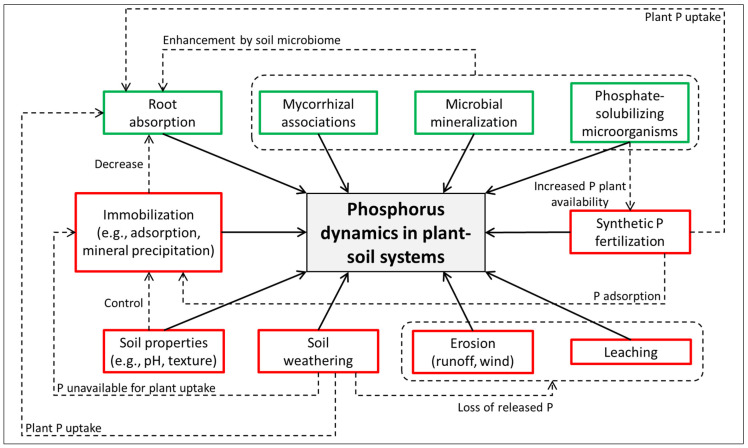
Overview of factors that influence (solid arrows) P dynamics in plant–soil systems. Biotic and abiotic factors are shown in green and red boxes, respectively. Dashed arrows show interactions between biotic and abiotic factors.

**Figure 3 biology-14-00433-f003:**
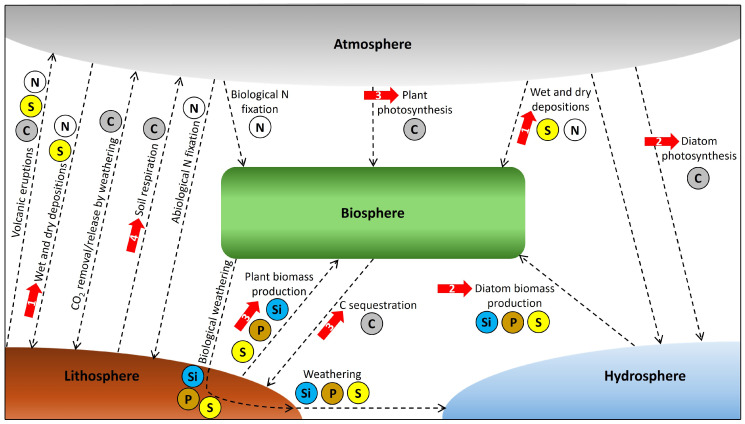
Schematic overview of natural key processes that link biogeochemical C, N, P, S, and Si cycles on a global scale. Dashed arrows indicate directional interactions between biogeochemical cycles such as photosynthesis, N fixation, weathering, or C sequestration, which connect the different Earth’s spheres (i.e., the litho-, hydro-, bio-, and atmosphere). Biogeochemical cycles that are mainly involved in these interactions are indicated by colored element symbols. Principle element flux directions are indicated by arrow directions. Red solid arrows show disturbances resulting from human activities. These disturbances are represented by numbers in the red arrows: 1 = fossil fuel combustion, 2 = eutrophication, 3 = land-use (agriculture, deforestation), and 4 = global warming. Note that the Earth’s spheres naturally are inextricably linked with each other and that the separated visualization in this figure is for illustration purposes only.

**Figure 4 biology-14-00433-f004:**
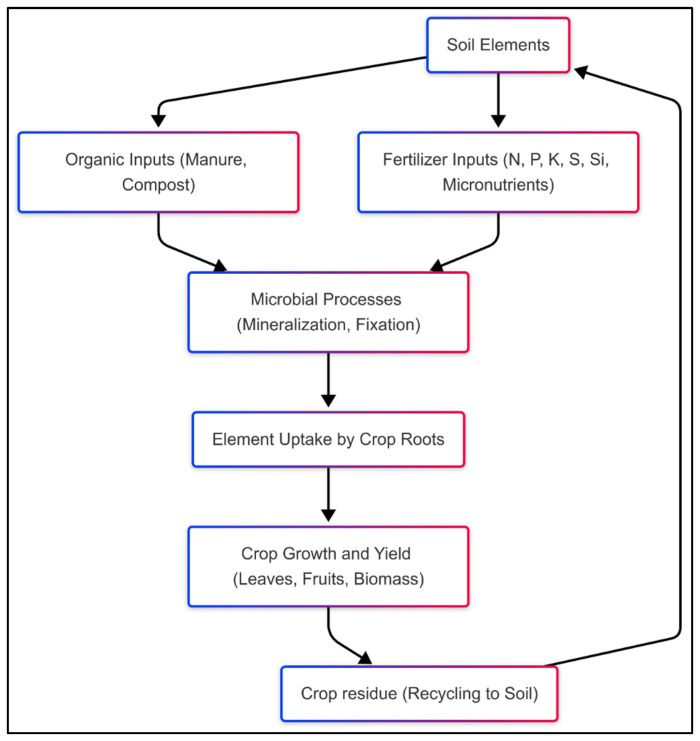
Nutrient/element transfer pathway from soil to crops.

**Figure 5 biology-14-00433-f005:**
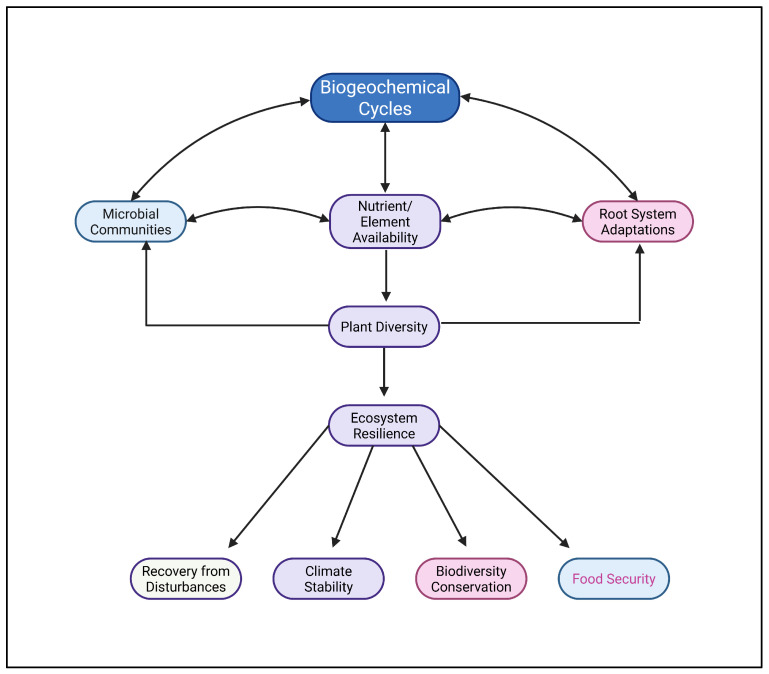
Interactions between biogeochemical cycles, plant biodiversity, and ecosystem resilience in plant–soil systems.

**Table 1 biology-14-00433-t001:** Comparative overview of N sources and their contribution to soil fertility.

Nitrogen Source	Soil Fertility Contribution	Crop Productivity Contribution	Advantages	Limitations	Reference
Synthetic Fertilizers	High immediate nutrient availability	High yield response	Quick release, tailored compositions	Risk of leaching, environmental harm	[53]
Manure	Slow nutrient release	Moderate to high, depending on quality	Organic matter improvement	Variable nutrient content	[54]
Compost	Slow and steady nutrient release	Moderate	Enhances soil structure	Requires time for production	[55]
Cover Crops	Long-term improvement	Indirect, through soil health	Erosion control, organic matter boost	Requires land-use during growth period	[56]
Leguminous Plants	Biological nitrogen fixation	High in compatible systems	Self-sustaining nitrogen source	Limited to suitable crops	[57]
Biofertilizers	Variable, depends on microbial activity	Variable	Environmentally friendly	Requires optimal conditions	[58]
Organic-Inorganic Mix	Balanced nutrient availability	High yield response	Improves nutrient-use efficiency	Complex management	[59]
High-Quality Organic Resources	Moderate to high	High, particularly in low-fertility soils	Reduces dependency on synthetic inputs	Requires high-quality material	[60]

**Table 2 biology-14-00433-t002:** Overview of S containing soil amendments and corresponding sources, effects on plant–soil systems, and advantages as well as limitations of application.

Soil Amendment	Source	Impact on Plant–Soil System	Advantages	Limitations	Reference
Elemental Sulfur	Naturally mined sulfur deposits	Lowers soil pH, improves nutrient availability	Long-term sulfur supply, pH adjustment	Requires microbial oxidation for effect	[55]
Gypsum (CaSO_4_·2H_2_O)	By-product of industrial processes or mined	Supplies calcium and sulfur; improves soil structure	Reduces aluminum toxicity in acidic soils	Limited to soils needing calcium	[54]
Ammonium Sulfate	By-product of fertilizer manufacturing	Rapid sulfur and nitrogen source	Quick nutrient release	Potential to acidify soils	[53]
Sulfur-Coated Urea	Industrially coated nitrogen fertilizer	Slow-release sulfur and nitrogen source	Provides consistent nutrient availability	Expensive to produce	[59]
Organic Matter	Plant residues, manure, compost	Gradual sulfur release through decomposition	Improves soil organic matter and soil fertility	Variable sulfur content	[58]
Biosolids	Treated sewage sludge	Supplies sulfur and organic matter	Recycling waste material	May contain “heavy metals” or contaminants	[81]
Potassium Sulfate	By-product of potash mining	Provides potassium and sulfur	Improves potassium levels	Limited to crops needing potassium	[82]
Sulfuric Acid	Industrial sulfur by-product	Lowers soil pH quickly in alkaline soils	Rapid correction of high soil pH	Risk of over-acidification	[57]

**Table 3 biology-14-00433-t003:** Information on biosilicification (Si pools and annual Si uptake rates) by testate amoebae, diatoms, and sponges in soils of terrestrial ecosystems. Note different units for Si pool sizes.

Year	Ecosystem	Organism	Si Pool Size	Biosilicification Rate	Reference
2007	Various forests	Testate amoebae	Up to 0.8 kg Si ha^−1^	Up to 106 kg Si ha^−1^ yr^−1^	[100]
2013	Beech forest ‘Beerenbusch’	”	1.9 kg Si ha^−1^	17 kg Si ha^−1^ yr^−1^	[101]
2014	Initial ecosystem states (different artificial catchments)	”	Up to 0.7 kg Si ha^−1^	Up to 16 kg Si ha^−1^ yr^−1^	[97]
2015	Various forests	”	Up to 4.7 kg Si ha^−1^	Up to 80 kg Si ha^−1^ yr^−1^	[94]
2016	Initial ecosystem states (artificial catchment ‘Chicken Creek’)	”	Up to 0.06 kg Si ha^−1^	--	[98]
”	”	Diatoms	Up to 0.3 kg Si ha^−1^	--	”
”	”	Sponges	Up to 0.2 kg Si ka^−1^	--	”
”	Various habitats in a nature reserve (artificial catchment ‘Mere Sands Wood’)	Testate amoebae	Up to 82 ng Si g^−1^ dm	--	[102]
”	”	Diatoms	Up to 58 ng Si g^−1^ dm	--	”
2017	Initial ecosystem states (artificial catchment ‘Chicken Creek’)	Testate amoebae	Up to 0.4 kg Si ha^−1^	--	[99]
”	”	Diatoms	Up to 1.6 kg Si ha^−1^	--	”
”	”	Sponges	Up to 0.5 kg Si ha^−1^	--	”
2020	Floodplain (Watarase retarding basin)	Testate amoebae	Up to 2.9 μg Si g^−1^ dm	--	[103]
”	”	Diatoms	Up to 12.8 μg Si g^−1^ dm	--	”
”	Peatland and cropland sites (Dajiuhu National Wetland Park)	Testate amoebae	Up to 5.3 μg Si g^−1^ dm	--	[104]
2021	Natural and cultivated *Sphagnum* sites	”	Up to 0.1 μg Si per 150 testate amoeba shells	--	[105]
2022	Various peatlands	”	Up to 97 ng Si per 150 testate amoeba shells	--	[106]

” = ditto; -- = no data available; dm = dry mass.

**Table 4 biology-14-00433-t004:** Key microbial species and their functional roles in nutrient cycling.

Microbial Species (Scientific Name)	Functional Role	Nutrient Cycling Process	Key Outputs/Impacts	Reference
*Azotobacter vinelandii*	Nitrogen fixation	Atmospheric N_2_ → Ammonia	Enhances soil nitrogen availability for plants.	[149]
*Bradyrhizobium japonicum*	Symbiotic nitrogen fixation	Forms nodules on legumes	Supplies nitrogen directly to host plants.	[150]
*Pseudomonas fluorescens*	Phosphate solubilization	Converts insoluble phosphorus	Increases bioavailability of phosphorus for plant uptake.	[151]
*Bacillus subtilis*	Phosphate solubilization	Organic phosphorus mineralization	Supports plant growth by enhancing soil phosphorus levels.	[152]
*Nitrosomonas europaea*	Nitrification	Ammonia → Nitrites	Facilitates conversion of nitrogen into usable forms, influencing nutrient cycling.	[153]
*Nitrobacter winogradskyi*	Nitrification	Nitrites → Nitrates	Ensures availability of nitrate for plant uptake but increases leaching risks.	[154]
*Paraburkholderia phytofirmans*	Plant growth promotion	Enhances phosphorus and nitrogen	Improves nutrient acquisition, fostering plant growth.	[155]
*Rhizobium leguminosarum*	Symbiotic nitrogen fixation	Forms nodules on legumes	Converts atmospheric nitrogen for host plants, improving soil fertility.	[156]
*Frankia* spp.	Nitrogen fixation in actinorhizal plants	Atmospheric N_2_ → Ammonia	Supports nitrogen levels in non-leguminous plants.	[157]
*Desulfovibrio desulfuricans*	Sulfate reduction	Sulfate (SO_4_^2−^) → Hydrogen sulfide (H_2_S)	Contributes to sulfur cycling in anaerobic environments, impacting soil and water chemistry.	[158]
*Thiobacillus thioparus*	Sulfur oxidation	Elemental sulfur → Sulfate (SO_4_^2−^)	Increases soil sulfate levels, promoting plant sulfur uptake.	[159]

**Table 5 biology-14-00433-t005:** Overview of soil management practices and corresponding C sequestration potentials.

Soil Management Practice	Description	Carbon Sequestration Potential	Impact on Soil Health	Reference
Conservation Tillage	Reduced tillage to minimize soil disturbance.	Moderate to high	Improves soil structure, reduces erosion, and enhances organic matter retention.	[149]
Cover Cropping	Planting cover crops during off-season periods.	High	Increases organic carbon inputs and reduces nutrient leaching.	[176]
Compost Addition	Application of compost to soils.	High	Enhances microbial activity, nutrient availability, and organic carbon.	[177]
Agroforestry	Integration of trees with agricultural crops.	Very high	Promotes biodiversity, reduces soil erosion, and increases carbon storage.	[178]
Biochar Amendment	Adding pyrolyzed biomass to soil.	High	Increases soil carbon stability, improves water retention, and supports microbial growth.	[179]
Crop straw recycling	Application of chopped straw to soil.	Moderate to high (long-term effects)	Replenishes plant available Si, reduces the need for N fertilizers, and increases organic carbon inputs in the long term.	[113]
Crop Rotation	Alternating crops to improve soil nutrient balance.	Moderate	Reduces pest buildup, enhances nitrogen use efficiency, and improves soil structure.	[154]
Integrated Livestock Management	Combining livestock and crop systems.	Moderate to high	Enhances nutrient recycling and boosts organic matter input through manure.	[155]
No-Tillage	Avoiding plowing entirely to maintain soil integrity.	High	Reduces erosion, improves water infiltration, and increases organic matter retention.	[156]
Perennial Grass Systems	Using perennial grasses for soil coverage.	Very high	Reduces erosion, improves soil structure, and enhances long-term carbon storage.	[157]

**Table 6 biology-14-00433-t006:** Overview of major anthropogenic threats and mitigation strategies.

Threat	Impact on Biogeochemical Cycles	Potential Mitigation Strategies	Reference
Excessive Fertilizer Use	Disrupts nitrogen and phosphorus cycles; causes eutrophication.	Precision agriculture, optimized fertilizer application, and crop-specific nutrient management.	[149]
Deforestation	Reduces carbon sequestration and alters nitrogen and silicon cycling.	Reforestation, afforestation, and agroforestry practices.	[198,205]
Desilication	Loss of Si from agricultural plant–soil systems.	Crop straw recycling, application of amorphous silica.	[111,113]
Industrial Pollution	Releases “heavy metals” and toxic compounds, affecting microbial activity and soil health.	Pollution control measures, phytoremediation, and stricter industrial regulations.	[206,207]
Urbanization	Alters land-use, leading to loss of soil organic matter and nutrient imbalances.	Urban green spaces, soil restoration projects, and sustainable urban planning.	[208]
Overgrazing by Livestock	Depletes soil nutrients and increases erosion, disrupting nutrient cycling.	Rotational grazing, controlled stocking rates, and land rehabilitation.	[209]
Waste Mismanagement	Accumulation of organic waste disrupts carbon and nitrogen cycles.	Composting, recycling, and waste-to-energy technologies.	[154]
Mining Activity	Causes soil degradation and disrupts phosphorus and sulfur cycles.	Land reclamation, sustainable mining practices, and ecosystem restoration.	[155]
Climate Change	Accelerates nutrient leaching and alters carbon, nitrogen, and water cycles.	Carbon capture technologies, renewable energy, and climate-smart agriculture.	[156]
Aquatic Pollution	Disturbs nutrient cycling in water bodies, leading to hypoxia.	Wetland restoration, buffer strips, and controlled effluent discharge.	[157]

## Data Availability

No data were used for the research described in the article.

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
