# Peer review of "Biogeochemical Cycles in Plant–Soil Systems: Significance for Agriculture, Interconnections, and Anthropogenic Disruptions"

_biology, 2025, doi:10.3390/biology14040433_

Round 1

Reviewer 1 Report

Comments and Suggestions for Authors

I have thoroughly reviewed the paper you submitted and am genuinely interested in your research. A comprehensive and systematic review was conducted on the biogeochemical cycles of key elements such as carbon, nitrogen, phosphorus, sulfur, and silicon in plant soil systems, covering the mechanisms, dynamics, interactions, and impacts on ecosystems of each element cycle, providing readers with rich knowledge and in-depth understanding. The extensive citation of classic research and the latest literature in related fields not only enhances the credibility and authority of the article, but also provides readers with rich reference resources for further in-depth research. Below, I will outline my observations and suggestions for enhancement.

  1. When exploring the cycles of various elements, specific details and examples of key processes can be further refined and supplemented.For example, in the carbon cycle, besides photosynthesis and microbial decomposition, respiration is also an important pathway. In addition, the author only briefly introduced the impact of land use, and there should be a lot of related carbon cycle content.
  2. In the nitrogen cycle, in addition to providing a general description of nitrogen fixation, nitrification, and denitrification, it is also possible to add discussions on the specific manifestations and differences of these processes under different types of soil and climatic conditions, as well as examples of their impacts on agricultural production and ecosystems.
  3. In section 2.3.1, the forms of phosphorus in soil can be further refined. In addition to the conversion of organic and inorganic phosphorus, different forms of inorganic phosphorus in soil (such as iron phosphate, aluminum phosphate, calcium phosphate, etc.) and their specific effects on phosphorus availability can also be mentioned. This will enable readers to have a more comprehensive understanding of the chemical behavior and availability of phosphorus in soil.
  4. The introduction of sulfur cycle should include different forms of sulfur and the utilization of sulfur in plant growth and farmland. Currently, the introduction of carbon, nitrogen, phosphorus, and sulfur cycle is too simplistic.
  5. Line 248 Why is this sentence bolded?
  6. When discussing the stability of ecosystems in nutrient poor soils, mechanisms such as microbial symbiosis and adaptive root systems were mentioned, but the specific connection between these mechanisms and the element cycling mentioned earlier was not clarified. It is suggested to further clarify how these mechanisms interact with element cycling to maintain ecosystem stability.
  7. The role of silicon in enhancing the resilience of soil microbial communities is a highlight. This viewpoint can be further expanded to explore the potential synergistic effects between silicon and other element cycles, as well as their comprehensive impact on ecosystem resilience, providing new directions for future research.
  8. The topic of the article is too large, but the content is not detailed enough.

Author Response

Response to the reviewers’ comments

 Reviewer #1

I have thoroughly reviewed the paper you submitted and am genuinely interested in your research. A comprehensive and systematic review was conducted on the biogeochemical cycles of key elements such as carbon, nitrogen, phosphorus, sulfur, and silicon in plant soil systems, covering the mechanisms, dynamics, interactions, and impacts on ecosystems of each element cycle, providing readers with rich knowledge and in-depth understanding. The extensive citation of classic research and the latest literature in related fields not only enhances the credibility and authority of the article, but also provides readers with rich reference resources for further in-depth research. Below, I will outline my observations and suggestions for enhancement.

Response: Thanks a lot for your insightful comments on our manuscript. We carefully revised our manuscript based on these comments (and the comments of Reviewer #2).

1. When exploring the cycles of various elements, specific details and examples of key processes can be further refined and supplemented. For example, in the carbon cycle, besides photosynthesis and microbial decomposition, respiration is also an important pathway. In addition, the author only briefly introduced the impact of land use, and there should be a lot of related carbon cycle content.

Response: We substantially reworked section 2 and added a third subsection to every element cycle description, which presents a short nutrient/element profile. In this context, information on carbon respiration and the impact of land-use on carbon cycling has been advanced (in the course of the revision we also substantially reworked Fig. 3).

2. In the nitrogen cycle, in addition to providing a general description of nitrogen fixation, nitrification, and denitrification, it is also possible to add discussions on the specific manifestations and differences of these processes under different types of soil and climatic conditions, as well as examples of their impacts on agricultural production and ecosystems.

Response: We reworked this section substantially and added more general information, e.g., on the nitrogen supply in agricultural plant-soil systems. However, as this section is thought to provide a (relatively) short overview of biogeochemical cycles, we did not add specific information on soil and/or climate conditions during revision. Nevertheless, during revision we added numerous new references, where the gentle reader can find much more information related to these fields.

3. In section 2.3.1, the forms of phosphorus in soil can be further refined. In addition to the conversion of organic and inorganic phosphorus, different forms of inorganic phosphorus in soil (such as iron phosphate, aluminum phosphate, calcium phosphate, etc.) and their specific effects on phosphorus availability can also be mentioned. This will enable readers to have a more comprehensive understanding of the chemical behavior and availability of phosphorus in soil.

Response: This section was revised substantially, and numerous new references were added during revision, which will allow the gentle reader to get into more detail, if interested. Detailed information on inorganic phosphorus in soil, for example, can be found in Hinsinger (2001) (new ref. no. 59) and Shen et al. (2011) (new ref. no. 60).

4. The introduction of sulfur cycle should include different forms of sulfur and the utilization of sulfur in plant growth and farmland. Currently, the introduction of carbon, nitrogen, phosphorus, and sulfur cycle is too simplistic.

Response: We substantially reworked the sulfur section during revision. Information on sulfur management in agricultural plant-soil systems can now be found in subsection 2.4.3. (incl. Table 2). In general, we carefully reworked every element section to provide a short examination of each cycle and selected element-specific aspects. During this revision numerous new references were added for further reading.

5. Line 248 Why is this sentence bolded?

Response: This sentence was bolded accidently. We corrected this mistake in the revised manuscript.

6. When discussing the stability of ecosystems in nutrient poor soils, mechanisms such as microbial symbiosis and adaptive root systems were mentioned, but the specific connection between these mechanisms and the element cycling mentioned earlier was not clarified. It is suggested to further clarify how these mechanisms interact with element cycling to maintain ecosystem stability.

Response: Good point. We added some more information (and corresponding references) in relation to this aspect to the revised version of our manuscript.

7. The role of silicon in enhancing the resilience of soil microbial communities is a highlight. This viewpoint can be further expanded to explore the potential synergistic effects between silicon and other element cycles, as well as their comprehensive impact on ecosystem resilience, providing new directions for future research.

Response: We added some more information and corresponding references dealing with this aspect to the revised version of our manuscript. Moreover, corresponding directions for future research were added. Thanks for the valuable hint.

8. The topic of the article is too large, but the content is not detailed enough.

Response: We substantially revised our manuscript based on your insightful comments. We are convinced that our manuscript broadly benefitted from this revision. Thanks again for your time and efforts.

Reviewer 2 Report

Comments and Suggestions for Authors

Several questions and remarks regarding the review by Zaman et al. titled “Interconnected Biogeochemical Cycles in Plant-Soil Systems: Mechanisms, Impacts, and Implications for Sustainable Land-Use Management”.

The biogeochemical cycles of nutrients such as nitrogen and carbon have been extensively investigated. Unlike the phosphorus and sulphur cycles. The disparities in the quantity of information provided in sections 2.1-2.4 serve as proof of this. Only two brief paragraphs are devoted to the phosphorus and sulphur cycles, which receive the least attention. More details and analyses of the biogeochemical cycles of phosphorus and sulfur are needed.

There is no unified logic in materials presentation. The carbon cycle has been defined, with a focus on soil carbon sequestration and land use influences on the carbon cycle. The nitrogen cycle is primarily described in terms of the mechanisms of nitrogen transformation in soil. The phosphorous cycle is restricted by defining its availability to plants and its management limitations. The sulphur cycle is confined to a summary of the element's significance in plants and a few words about the sulphur transformation activities. The review would be greatly improved if there were a unified presentation logic. For example:

  1. availability of nutrient in soils
  2. nutrient in plant nutrition
  3. nutrient transformation processes
  4. limitations and management
  5. impact of land-use changes on nutrient cycling

Figure 2. Colour differences in the lines (purple, blue, red, orange, yellow, red) have any relevance?

Author Response

Response to the reviewers’ comments

Reviewer #2

Several questions and remarks regarding the review by Zaman et al. titled “Interconnected Biogeochemical Cycles in Plant-Soil Systems: Mechanisms, Impacts, and Implications for Sustainable Land-Use Management”.

Response: Thanks a lot for your insightful comments on our manuscript, which really helped us to improve its overall quality.

1. The biogeochemical cycles of nutrients such as nitrogen and carbon have been extensively investigated. Unlike the phosphorus and sulphur cycles. The disparities in the quantity of information provided in sections 2.1-2.4 serve as proof of this. Only two brief paragraphs are devoted to the phosphorus and sulphur cycles, which receive the least attention. More details and analyses of the biogeochemical cycles of phosphorus and sulfur are needed.

Response: You are right. We substantially revised section 2 to meet your concerns. In this context, we divided every element section into three subsections following your suggestion below. In doing so, much more information and numerous references for further reading were added to all element sections in the revised version of our manuscript.

2. There is no unified logic in materials presentation. The carbon cycle has been defined, with a focus on soil carbon sequestration and land use influences on the carbon cycle. The nitrogen cycle is primarily described in terms of the mechanisms of nitrogen transformation in soil. The phosphorous cycle is restricted by defining its availability to plants and its management limitations. The sulphur cycle is confined to a summary of the element's significance in plants and a few words about the sulphur transformation activities. The review would be greatly improved if there were a unified presentation logic. For example:

  1. availability of nutrient in soils
  2. nutrient in plant nutrition
  3. nutrient transformation processes
  4. limitations and management
  5. impact of land-use changes on nutrient cycling

Response: Good point, we really like the idea of this way of presentation. However, to keep the overall structure of our review article, we decided to apply this outline to section 2 only. In this context, we added a new subsection to every element section, which now briefly provides information on the element’s natural plant availability, its significance in plant nutrition, molecular transformation processes, element limitations and management, and consequences of land-use for the biogeochemical cycle of this element. Selected aspects from the first subsection are then presented in the following subsections in more detail. Thanks a lot for this insightful recommendation.

3. Figure 2. Colour differences in the lines (purple, blue, red, orange, yellow, red) have any relevance?

Response: We entirely reworked this figure and hope it is much clearer now. Thanks again for your comments on our manuscript. We really appreciate your efforts.

Round 2

Reviewer 1 Report

Comments and Suggestions for Authors

Accept

Reviewer 2 Report

Comments and Suggestions for Authors

All the issues I cared about were addressed by the authors.  The manuscript has been carefully revised.